# The Use of PediSTAT Application by Paramedics Working in Saudi Arabia to Reduce the Risk of Medication Error for Pediatric Patients

**DOI:** 10.3390/pediatric17010009

**Published:** 2025-01-16

**Authors:** Nesrin Alharthy, Raghad Abuhaimed, Munirah Alturki, Shatha Alanazi, Raghad Althaqeb, Alanowd Alghaith, Abdullah Alshibani

**Affiliations:** 1Pediatrics Emergency Department, King Abdulaziz Medical City, Riyadh 14611, Saudi Arabia; 2Emergency Medical Services Department, College of Applied Medical Sciences, King Saud bin Abdulaziz University for Health Sciences, Riyadh 11481, Saudi Arabia; raghad1abuhaimed@gmail.com (R.A.); munirah.n.a@outlook.com (M.A.); shatha.al0041@gmail.com (S.A.); althaqebksau@gmail.com (R.A.); alghaitha01@gmail.com (A.A.); alshibania@ksau-hs.edu.sa (A.A.); 3King Abdullah International Medical Research Center, Riyadh 11481, Saudi Arabia

**Keywords:** paediatric, durg, dose, phone, prehospital, ambulance, emergency

## Abstract

Background/Objectives: This study aimed to assess and compare the rates of medication error (ME) using the PediSTAT application compared to the conventional method of calculating the correct dose and determining the appropriate route of medication administration for common pediatric emergencies. Methods: A prospective cross-sectional study design was used for the study. Data were collected using a questionnaire that was distributed to certified paramedics holding a bachelor’s degrees or higher and working in Riyadh City, Saudi Arabia. Alternate simple random sampling was used to recruit the participants into two groups using the same questionnaire: the PediSTAT group and the conventional method group. The questionnaire contained four pediatric emergency vignettes: cardiac arrest, asthma exacerbation, seizures, and hypoglycemia. Results: A total of 63 participants agreed to the study. Almost 80% of them were males, 81% held bachelor’s degrees, and 87% were certified in pediatric resuscitation courses. The findings of the study showed that the use of the PediSTAT application increased accuracy and reduced the risk of ME for common pediatric emergencies. This was shown to be statistically significant for asthma medication dose (*p*-value < 0.001, 95% CI 0.034–0.352), midazolam dose (*p*-value = 0.012, 95% CI 0.030–0.764), and hypoglycemia medication dose (*p*-value < 0.001, 95% CI 0.046, 0.452). Conclusions: The study findings supported the use of standardized precalculated applications such as PediSTAT, which was shown to reduce the risk of ME in prehospital care for pediatric emergencies.

## 1. Introduction

Medication error (ME) is defined as any preventable incident that results in misusing pharmaceuticals and causing harm to the consumer [1,2]. It is a life-threatening condition that can result in an adverse drug event (ADE), which is a medication-related injury that can result in morbidity and mortality [3]. The World Health Organization (WHO) has continuously aimed and called to reduce the severe, serious, and avoidable ME harm that can be introduced by healthcare professionals [4].

Emergency medical services (EMS) personnel are healthcare professionals who are responsible for responding to all out-of-hospital emergency cases of patients from all age groups. Working in such emergency care settings, which are known as highly stressful healthcare settings, could increase the risk of ME by EMS personnel [5,6]. Managing the critical cases of pediatric patients in such settings is even more stressful and challenging since medications for pediatrics must be calculated based on the patient’s weight, which may result in arithmetic errors, potentially increasing the risk of ME. In Saudi Arabia, 3.4% of emergency cases requiring out-of-hospital care are related to pediatric patients (aged < 15 years) [7].

The healthcare system in Saudi Arabia primarily consists of public and private hospitals. Public hospitals are mainly run by and follow the guidelines of the Ministry of Health in Saudi Arabia. Private hospitals are run by their board but follow the guidelines of the Ministry of Health. Both public and private hospitals are equipped and prepared with personnel, equipment, and facilities to handle pediatric patients.

Evidence from Saudi Arabia showed that out of 109,382 prescribed medications over a two-year period to pediatric patients at a large tertiary hospital, 9123 (8.3%) MEs were reported [8]. These MEs were related to 84 different types of medications [8]. Of these ME cases, wrong frequency was the most common type of error (39.1%) [8]. Other common types of errors included the wrong drug (12.5%), wrong concentration or strength (12.4%), and wrong dose (11.1%) [8]. Unfortunately, there is a lack of available data regarding ME by EMS personnel working in Saudi Arabia. Internationally, drug dosing errors for pediatric patients administered by EMS personnel working in out-of-hospital settings were at a high rate despite the implementation of pediatric drug dosing references [9]. Such errors were related to overdoses, underdoses, and dilution errors [9]. Although this evidence showed high rates of ME among EMS personnel, it might not be applicable to paramedics working in Saudi Arabia due to their professional qualifications and scope of practice. ME related to medication administration and dosing can be severely harmful or fatal [10], which highlights the need to reduce the risk of MEs to prevent these adverse outcomes.

To reduce the risk of ME, the literature examined the use of standardized precalculated pediatric doses for out-of-hospital emergencies [6,11]. Standardized and precalculated pediatric medication doses were shown to significantly reduce ME and increase paramedics’ confidence in pediatric medication dosing [12]. Furthermore, a literature review showed that using pediatric height-based drug doses reduced MEs from (22%) to (9.9%), supporting the need to use standardized measures for pediatric medication dosing and administration to reduce MEs in out-of-hospital settings [13]. With the advancement of technology and the availability of smartphones and applications, multiple software applications were reported to effectively reduce pediatric MEs during out-of-hospital emergency care [14,15,16]. One of the most common smartphone applications to achieve the goal of reducing MEs for pediatric patients is PediSTAT, which is a tool intended to assist healthcare professionals mainly with drug dosing for pediatric patients using patient weight. To our knowledge, no study has specifically assessed the use of PediSTAT among paramedics nationally. Therefore, this study aimed to assess and compare the rates of MEs using the smartphone application (PediSTAT) and the conventional method of choosing correct routes and providing correct doses using pediatric clinical vignettes.

## 2. Materials and Methods

### 2.1. Study Design

A descriptive cross-sectional design was used in this study to assess and compare the rates of MEs when using PediSTAT compared to the conventional method of choosing the correct route and dose of medications for pediatric patients. A validated questionnaire involving pediatric clinical vignettes was distributed to our study sample in Riyadh City, Saudi Arabia, from 1 September 2022 to 28 February 2023. Ethical approval was obtained from the King Abdullah International Medical Research Center, Riyadh, Saudi Arabia, (approval number: IRB/1603/22) on 11 August 2022.

### 2.2. Setting

Data were collected from paramedics working in EMS in Riyadh City, Saudi Arabia. Riyadh City is the capital of Saudi Arabia with a total population of around eight million people. Several EMS systems are currently running in the city to provide prehospital care for those in need. Saudi Red Crescent Authority is the main EMS system in Saudi Arabia to provide prehospital care for people living in the kingdom, which includes Riyadh City. Other EMS systems in the city include hospital-based EMS systems under the Ministry of Health, which mainly respond to both emergency and low-equity cases in and outside of the hospital, along with critical care transfers. Private EMS systems are also present and work under private hospitals to cover interfacility transfers to these hospitals and respond to emergency cases in large events and festivals conducted in the city.

### 2.3. Survey Administration and Content

An online survey using Google Forms was used to collect responses from the participants in the English language. An informed consent form was provided to all participants prior to completing the survey. The survey collected demographic information from the participants. It also included several pediatric clinical vignettes in each of which the participants reported the correct route and dose of the medication.

Due to the lack of a previously validated questionnaire to assess the use of PediSTAT compared to the conventional methods to correctly identify the route and dose of emergency medication for pediatric patients, a questionnaire was developed for this study. When developing the questionnaire, the aim of the study and the targeted population were considered. Pediatric clinical vignettes, in each of which the participants were asked about the route and dose of medication deemed appropriate, were used to maintain objectivity and achieve the study's aims to assess the use of PediSTAT and compare it with the conventional method. Demographic information was collected to describe the participants and identify factors associated with the use of either PediSTAT or the conventional method.

The questionnaire was piloted as randomly selected researchers and paramedics were invited to complete the survey (two paramedic researchers holding a PhD degree, one pediatric emergency medicine consultant, and three paramedics). Face validity was followed by the research team to approve the questionnaire before dissemination to the participants. The questionnaire was initially sent to the invited researchers and clinicians. A subsequent meeting was conducted with the research team to receive their feedback about the questionnaire. The aim of this pilot was to assess the flow and clarity of the questions and clinical vignettes, and the questionnaire supported the achievement of the study’s aims. The piloting process showed that no major changes were required.

The questionnaire has two main parts: demographic information and clinical vignettes. The collected demographic information included age in years, gender, degree, name of higher education program (if applicable), workplace, specialized pediatric care course, and the name of the course (if applicable). For the clinical vignettes, four vignettes (cardiac arrest, severe asthma, seizure, and hypoglycemia) were provided to the participants, and they were asked to report the route and dose of the medication in each vignette. A copy of the questionnaire is provided in the Appendix A.

### 2.4. Participation and Recruitment

The targeted population in this study is paramedics working in Riyadh City, Saudi Arabia. The prehospital care system in Saudi Arabia is predominantly run by paramedics and emergency medical technicians. Paramedics usually hold at least a bachelor’s degree in EMS or a higher education degree (master’s degree or Doctor of Philosophy degree). The scope of work for paramedics working in Saudi Arabia primarily involves emergency response to cases in the prehospital care settings and/or inter-facility and intra-facility critical care transfers. The involvement of paramedics in fire stations is still not well developed. Therefore, the invited paramedics in this study were working either in hospitals or the Saudi Red Crescent Authority, which is the main authority in Saudi Arabia to respond to all emergency cases in prehospital care settings. Emergency medical technicians were excluded from this study as they are not allowed to provide medications to all patients. Medical doctors and nurses were also excluded from this study as they do not usually work in prehospital care in Saudi Arabia.

An online invitation outlining the study’s aim and objectives and how paramedics could participate in the study was sent through social media platforms (WhatsApp and Telegram) among workplace and academic groups. The participants who accepted the invitation and wanted to participate in the study were divided into two groups randomly (1:1): the control group (conventional method) and the intervention group (PediSTAT method). The method of randomization was conducted using Excel sheets and those who agreed to participate were given a unique ID number. These ID numbers were then entered into an Excel sheet to randomize the participants into either the intervention or control group. Both groups were provided with the same questionnaire and no additional information was obtained from either group. The participants from both groups were individually contacted and provided with the link to complete the survey. For the control group, each participant was instructed to complete the questionnaire using the conventional method that they usually use and without the use of any smartphone application that identifies the route and calculates the dose of medications for pediatric patients, including PediSTAT. The control group declared no use of any smartphone application when completing the survey. For the intervention group, the participants were asked to complete the questionnaire using the PediSTAT application only on their smartphones, whether they already had the application or had to install it on their smartphones. If any of them did not have the application or did not wish to have it on their smartphones, a member of the research team arranged a time that was the best for both of them for the researcher to visit the participant in his/her workplace and provide the application on the research team member’s smartphone. The completion of the survey for each participant was monitored by one of the study team members. The participant was provided with the link to the online questionnaire at a set time and the participant was asked to complete the questionnaire within one hour of receiving the link. A single submission only was available for all participants. The study team members reported no issues that any of the participants in the study exceeded the one-hour window for completing the survey.

### 2.5. Intervention

PediSTAT application was the used intervention in this study. It is an application tool that can be installed on smartphones and used promptly. It calculates medication doses for pediatric patients based on age, weight, length, and Broselow tape color codes. PediSTAT is owned and operated by James M. Kempema and permission was obtained by the owner. The research team used the application in accordance with the application licensing protocol (for more details on the PediSTAT app please refer to Appendix A).

### 2.6. Data Analysis

All collected responses were converted into an Excel spreadsheet and then coded for analysis. Data analysis was performed using IBM SPSS statistics version 28 (IBM, New York, NY, USA). Descriptive analysis was performed using median and interquartile ranges for continuous variables and frequency and percentages for categorical variables. Results were displayed in tables and figures. For the responses to clinical vignettes, the choice of the medication route was either the correct answer, if the predetermined correct route was chosen, or the wrong answer if other incorrect routes were chosen (i.e., medication route error). The medication dose was predetermined, and if the dose was the same or within 10% above or lower than the predetermined correct dose it was considered the correct answer. If it exceeded 10% above or lower than the predetermined correct dose, it was considered a wrong answer (i.e., medication dose error).

Pearson chi-square and Fisher exact tests were used to compare the correct and wrong answers of the clinical vignettes and to compare whether the certification of pediatric care courses was significantly associated with the ME rate for each method applied (conventional versus PediSTAT). A *p*-value < 0.05 was predetermined to be considered statistically significant.

## 3. Results

### 3.1. Characteristics of the Study Participants

A total of 63 participants out of 120 (response rate 52.5%) agreed to participate in the study and completed the survey (Figure 1). The participants were randomly assigned to two groups: 32 in the conventional method group and 31 in the PediSTAT group (Figure 1). The median age of the participants was 27 years (IQR 25, 32), and most participants were males (79.4%). Most participants held a bachelor’s degree (81%) and were certified in pediatric life support courses (87.3%) (Table 1). The baseline information of the participants in each group is shown in Table 1. There were no significant statistical differences in the baseline information between the control and intervention groups.

### 3.2. Findings from the Control Group

The ability of paramedics to administer pediatric medication doses and routes using conventional methods and the PediSTAT EMS application was evaluated, as shown in Table 2. In the cardiac arrest scenario, those who used conventional methods had a 71.9% accuracy rate for the epinephrine dose, while only 53.1% were correct with the epinephrine route. For the asthma scenario, the accuracy rate for medication choice was 56.2%, medication dose was 18.8%, and medication route was 59.4%. The seizure scenario had a midazolam dose accuracy rate of 68.8%, while the route had a 93.8% accuracy rate. Lastly, in the hypoglycemia scenario, there was a 93.8% accuracy rate for medication choice, 37.5% for medication dose, and 96.9% for medication route.

### 3.3. Findings from the Interventional Group

For PediSTAT users, the results showed that those who used the application had a 90.3% accuracy rate for the epinephrine dose in the cardiac arrest scenario. Only 74.2% were correct with the epinephrine route. For the asthma scenario, the accuracy rate for medication choice was 67.7%, medication dose was 67.7%, and medication route was 67.7%. In the seizure scenario, the midazolam dose accuracy rate was 93.5%, while the route had a 100.0% accuracy rate. In the hypoglycemia scenario, there was a 93.5% accuracy rate for medication choice, 80.6% for medication dose, and 96.8% for medication route.

### 3.4. Comparison Between the Two Groups

The use of the PediSTAT application has shown an increase in the accuracy of answers, especially in asthma medication dose, with a *p*-value of 0.001 (95% CI 0.034–0.352), midazolam dose, with a *p*-value of 0.012 (95% CI 0.030–0.764), and hypoglycemia medication dose, with a *p*-value of 0.001 (95% CI 0.046, 0.452).

The comparison of the wrong answers between the PediSTAT users and conventional method users is presented in Figure 2. In the cardiac arrest cases, the rates of incorrect dose and route of epinephrine in the PediSTAT group were lower than those reported in the conventional method group (Table 2). The same is true for the asthma cases for medication choice, dose, and route, as those who used PediSTAT reported lower rates of wrong answers (Table 2). Indeed, the dose of the asthma medication was significantly lower in the PediSTAT group (*p*-value < 0.001). For the seizure cases, the midazolam dose was significantly lower in the PediSTAT group (*p*-value = 0.012). However, the midazolam route was lower in the PediSTAT group, but this was not significant (*p*-value = 0.492). The hypoglycemia cases showed similar rates of wrong answers regarding medication choice and route between the PediSTAT and conventional groups except for the medication dose, which was significantly lower in the PediSTAT group (*p*-value ≤ 0.001) (Table 2).

### 3.5. Association Between Pediatric Life Support Certification and Accuracy of Answers

Paramedics with prior pediatric life support certification incorrect answers are presented in Table 3. The findings showed lower rates of medication errors regarding the choice, dose, and route of each medication for those who used the PediSTAT compared to the conventional group. However, it was not statistically significant at the *p*-value level of 0.05 (Table 3).

## 4. Discussion

This study assessed the accuracy and rate of medication error when using the PediSTAT application and compared it with the conventional methods for choosing the correct dose and route of administration of common emergency medications. The findings of the study showed that the rates of medication error rates were lower when using the PediSTAT application than with the conventional method for choosing the correct medication dose and route of administration. Furthermore, the use of the PediSTAT application was significantly associated with lower rates of medication error for asthma medication dose (*p*-value < 0.001), midazolam dose (*p*-value = 0.012), and hypoglycemia medication dose (*p*-value < 0.001). Moreover, sub-group analysis for paramedics who had prior pediatric life support certification showed that the use of the PediSTAT application resulted in lower rates of medication error than the conventional methods, but this was not statistically significant. Overall, the findings of the study showed that the use of the PediSTAT application can improve the accuracy and reduce the rates of medication error in prehospital care.

Evidence from available literature argues that pediatric resuscitation care is challenging in the prehospital setting and could impact outcomes and quality of resuscitation including the use of medications [17,18,19]. Medication dosing and safety in these critical circumstances represent a major challenge in the calculation via standard mathematical dosing and could impose the risk of medication error [20,21]. The literature reported that standardizing medication calculation using weight estimation methods impacted the medication error rate [22,23]. Standardized medication formulary and precalculated weight-based impact the rate of medication dosing error [12,24].

An open-label, simulation-based, multicenter, randomized clinical trial was conducted at 14 urban EMS centers in Switzerland with the aim of assessing whether the use of evidence-based mobile applications is associated with lower rates of medication error for pediatric patients in prehospital care as compared to the conventional method [15]. The findings from this trial showed that the use of mobile applications, in comparison with the conventional method, was significantly associated with decreased rates of medication error and time to drug delivery of emergency medication preparation in prehospital care [15]. Dedicated mobile apps have the potential to improve medication safety and change practices in pediatric prehospital care [15]. These findings are similar to our findings showing that the use of mobile applications is associated with improved accuracy and lower rates of medication error compared to the conventional methods of calculating the dose and choosing the route of emergency medications. Another study showed that the use of a mobile application (PedAMINES) was associated with lower rates of medication error and reduced time to administer drugs in emergency care settings [16], consistent with our findings. Overall, the available literature highlighted that the use of different mobile applications to calculate emergency medications for pediatric patients was associated with lower rates of medication error than the conventional methods and is consistent with our study findings.

The findings of our study showed that even when using PediSTAT, one-third of the answers seemed to be wrong. If this is accurate, then a significant amount of ME is occurring despite the use of smartphone applications. We are unsure of the reasons leading to the high rates of ME in this study when using the PediSTAT application. The aim of this study was to assess and compare the use of a smartphone application (PediSTAT) with the conventional method of calculating medication dose and determining the route of administration for pediatric patients by EMS personnel. We did not investigate why the participants gave wrong answers, whether they were in the control group or the intervention group. However, one potential reason for giving wrong answers in the PediSTAT group is unfamiliarity with the application.

Our study has notable strengths that contribute to the validity of the findings. Firstly, the selection of the participants in the PediSTAT and conventional method groups was random, which improved the internal validity of our findings and minimized selection bias. Furthermore, piloting the questionnaire added validity to our questionnaire and study findings as it allowed us to identify and address any issues or ambiguities in the questionnaire before formally distributing it to the participants. However, this study has some limitations that need to be highlighted. At first, we invited paramedics from Riyadh City only, as it is the capital city of Saudi Arabia and the most populated city in the country. This, however, could impact the generalizability of the study findings to the country of Saudi Arabia. Furthermore, we had a low response rate from paramedics as only 63 paramedics agreed to participate in this study. This could impact our findings about the benefit of using the PediSTAT application over the conventional methods in calculating the medication dose and choosing the medication route for pediatric patients in prehospital care. Moreover, some of the participating paramedics in the PediSTAT group were unfamiliar with the application and had never used it before. This could potentially impact the findings of this study although none of them reported any issues when using the application. Another important limitation that needs to be highlighted is that the results from written tests and actual behavior differ. Even if the participants recognized the correct answers, writing errors might occur. Also, even if they gave the correct answer in a test, they may undertake the wrong action. This intervention may be effective in preventing ME due to a lack of knowledge or calculation errors. However, it may not solve the already mentioned types of ME in the introduction and discussion section of this paper. In addition, using the PediSTAT application may introduce a risk of commercial bias. However, this study has no intention for any commercial purposes. It reports on the findings of using the conventional method compared to the smartphone application (PediSTAT was used as an example of multiple available smartphone applications) for calculating medication dose and determining the route of medication administration for pediatric patients by EMS personnel. We chose the PediSTAT application over other available smartphone applications because it resembles the length-based weight estimation Breslow tape, which is familiar to paramedics in Saudi Arabia.

## 5. Conclusions

The findings of this study revealed that the use of the PediSTAT application could significantly reduce the rates of medication error and improve the accuracy of determining the required medication dose and appropriate medication route of emergency medications for pediatric patients in prehospital care in Saudi Arabia. Pediatric life support certification was associated with lower, but not significantly lower, rates of medication error in the PediSTAT group compared to the conventional method of choosing the dose and route of medication for pediatric patients in prehospital care. The findings of this study highlighted the need to routinely use standardized precalculated applications including, for example, PediSTAT, as they could reduce the risk of medication error for pediatric patients and have the potential to save precious time that is usually needed in the prehospital care setting. Further larger-scale studies with larger sample sizes and real-time scenarios are needed to assess the use of PediSTAT in prehospital care. Other studies could investigate the use of other smartphone applications for pre-calculated doses and the route of administration of pediatric emergency medications for possible implementation in prehospital care.

## Figures and Tables

**Figure 1 pediatrrep-17-00009-f001:**
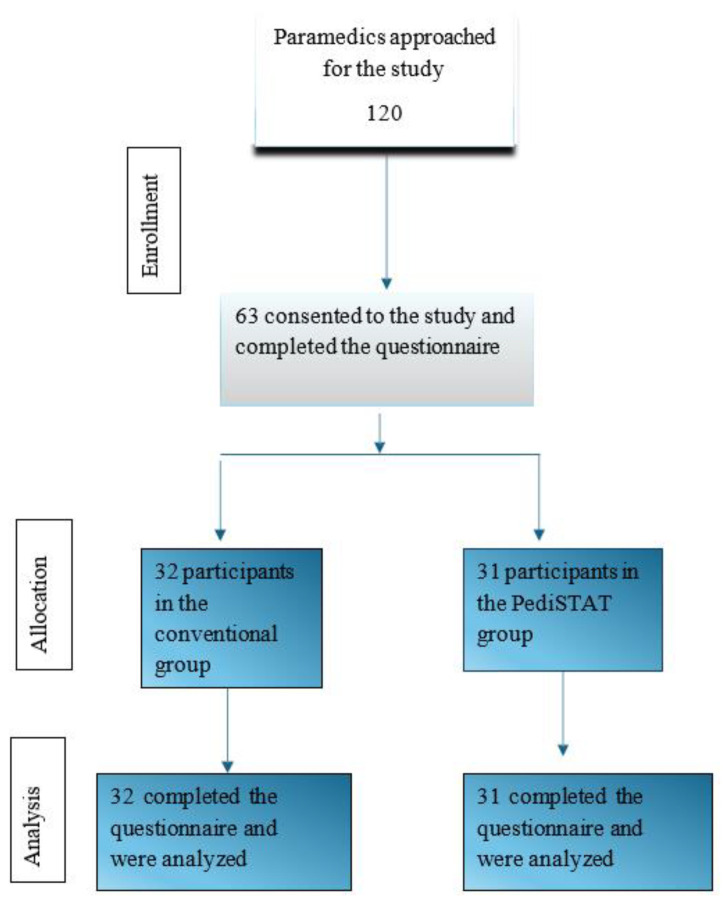
The study participants’ random allocations to conventional method or PediSTAT group.

**Figure 2 pediatrrep-17-00009-f002:**
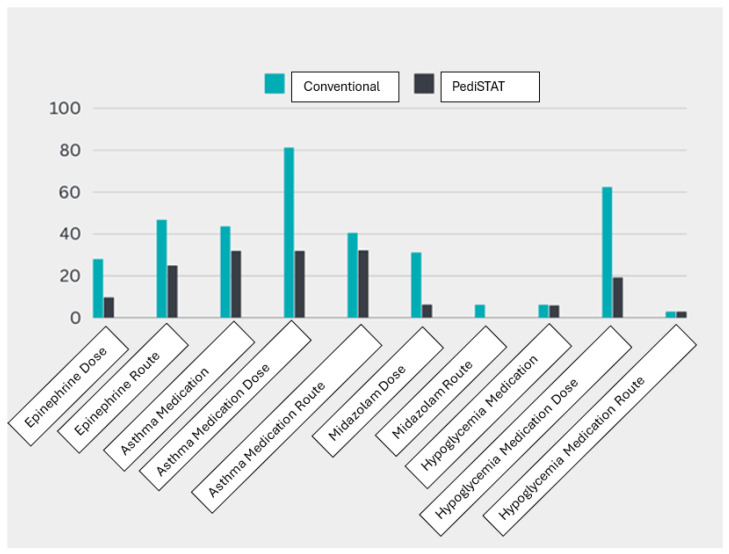
Comparing the error percentage by conventional method and PediSTAT method.

**Table 1 pediatrrep-17-00009-t001:** Demographic details of study participants.

Variable	Conventional (n = 32)	PediSTAT (n = 31)	Total (n = 63)
	Descriptive Statistics *		
**Gender**			
Males	26 (81.2)	24 (77.4)	50 (79.4)
Females	6 (18.8)	7 (22.6)	13 (20.6)
**Age in years**	28.5 (25, 34.5)	27 (25, 30)	27 (25, 32)
**Degree attained**			
Bachelor’s	25 (78.1)	26 (83.9)	51 (81)
Master’s	6 (18.8)	4 (12.9)	10 (15.9)
PhD	1 (3.1)	1 (3.2)	2 (3.2)
**Workplace**			
MNGHA	25 (78.1)	25 (80.6)	50 (79.4)
SRCA	7 (21.9)	6 (19.4)	13 (20.6)
**Having certification in any pediatric course**			
Yes	26 (81.2)	29 (93.5)	55 (87.3)
No	6 (18.8)	2 (6.5)	8 (12.7)

* Categorical variables, frequency, and percentage.

**Table 2 pediatrrep-17-00009-t002:** Comparing the correct and wrong answers by the two methods with *p*-value, odds ratio (OR), and 95% confidence interval (CI).

Particulars		Method Frequency (%)		OR (95% CI)	Chi-Square Value (df)	*p*-Value
		Conventional Method	PediSTAT EMS			
Epinephrine Dose	Correct Answers	23 (71.9)	28 (90.3)	0.274 (0.066, 1.131)	3.475 (1)	0.062
	Wrong Answers	9 (28.1)	3 (9.7)			
Epinephrine Route	Correct Answers	17 (53.1)	23 (74.2)	0.394 (0.136, 1.141)	3.015 (1)	0.082
	Wrong Answers	15 (46.90)	8 (25.8)			
Asthma Medication	Correct Answers	18 (56.20)	21 (67.7)	0.612 (0.219, 1.710)	0.882 (1)	0.348
	Wrong Answers	14 (43.8)	10 (32.3)			
Asthma Medication Dose	Correct Answers	6 (18.8)	21 (67.7)	0.110 (0.034, 0.352)	15.432 (1)	0.001 *
	Wrong Answers	26 (81.2)	10 (32.3)			
Asthma Medication Route	Correct Answers	19 (59.40)	21 (67.7)	0.696 (0.248, 1.953)	0.476 (1)	0.490
	Wrong Answers	13 (40.6)	10 (32.3)			
Midazolam Dose	Correct Answers	22 (68.80)	29 (93.5)	0.152 (0.030, 0.764)	6.280 (1)	0.012 *
	Wrong Answers	10 (31.20)	2 (6.5)			
Midazolam Route	Correct Answers	30 (93.8)	31 (100.00)	0.492 (0.381, 0.635)	2.001 (1)	0.492
	Wrong Answers	2 (6.2)	0 (0.0)			
Hypoglycemia Medication	Correct Answers	30 (93.8)	29 (93.5)	1.034 (0.136, 7.840)	0.001 (1)	1.000
	Wrong Answers	2 (6.20)	2 (6.5)			
Hypoglycemia Medication Dose	Correct Answers	12 (37.5)	25 (80.6)	0.144 (0.046, 0.452)	12.093 (1)	0.001 *
	Wrong Answers	20 (62.5)	6 (19.40)			
Hypoglycemia Medication Route	Correct Answers	31 (96.9)	30 (96.8)	1.033 (0.062, 17.282)	0.001 (1)	1.000
	Wrong Answers	1 (3.1)	1 (3.2)			

* Statistically significant at 5%. Test used is Pearson chi-square and Fisher exact test; df is degrees of freedom.

**Table 3 pediatrrep-17-00009-t003:** Comparing the error rates by the two methods regarding the certification of pediatric life support courses taken by the subjects.

Particulars	Method Frequency (%) *			*p*-Value
	Conventional	PediSTAT	Total	
Epinephrine dose	6 (66.7)	3 (33.3)	9 (100)	0.509
Epinephrine route	14 (70)	6 (30)	20 (100)	0.269
Asthma medication	11 (55)	9 (45)	20 (100)	0.615
Asthma medication dose	20 (66.7)	10 (33.3)	30 (100)	0.517
Asthma medication route	11 (52.4)	10 (47.6)	21 (100)	0.486
Midazolam dose	6 (85.7)	1 (14.3)	7 (100)	1.000
Midazolam route	2 (100)	0	2 (100)	-
Hypoglycemia medication	1 (33.3)	2 (66.7)	3 (100)	1.000
Hypoglycemia medication dose	15 (78.9)	4 (21.1)	19 (100)	1.000
Hypoglycemia medication route	1 (50)	1 (50)	2 (100)	-

* Frequency (%) of error rate in those who took the pediatric life support course by the two methods. Statistical test used was Fisher exact test at significance level 5%.

## Data Availability

Data will be made available upon reasonable request to the principal investigator of the study.

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
