# Peer review of "The Use of PediSTAT Application by Paramedics Working in Saudi Arabia to Reduce the Risk of Medication Error for Pediatric Patients"

_pediatrrep, 2025, doi:10.3390/pediatric17010009_

Round 1

Reviewer 1 Report (Previous Reviewer 3)

Comments and Suggestions for Authors Thank you for sending the revised manuscript. I confirmed the revision and all have completely acceptable answers for me. Thank you for your efforts.

Author Response

Reviewer 2 Report (Previous Reviewer 2)

Comments and Suggestions for Authors

Dear Authors, 

The manuscript has been positively improved. 

The study's limitation to PediSTAT after its work in Saudi Arabia should be highlighted in the title. I suggest emphasis on Saudi Arabia. The authors have added some content in the introductory section. Still, it would need a sentence about the context of the health service workflow, public versus private in the paediatric context. I highlight that the commercial bias of PediSata should be clarified, so I suggest justifying its use in the population sample and adding the authors' rational option.

Author Response

This manuscript is a resubmission of an earlier submission. The following is a list of the peer review reports and author responses from that submission.

Round 1

Reviewer 1 Report

Comments and Suggestions for Authors

Thank you for giving me the opportunity to evaluate such a great research paper.

The issue of drug misuse in prehospital emergency care is constantly being raised internationally. This paper can be used as a good resource for improving the quality of emergency care.

It is expected to be used as a good paper by explaining the latest paper review and research direction well.

It would be better for readers to understand if there was a brief introduction to Saudi Arabia's emergency medical system.

I would like to express my deepest encouragement to the researcher who has conducted such a difficult study.

 I would like to mention a few things about the paper that reviewed.

1. page 2, line 84

It would be helpful for readers to understand if there was a picture explaining the process of the experimental process.(Step1 – Step2 – Step3)

Or, it would be helpful to explain through a picture comparison of how to use the conventional method and the application.

line 96

The contents of the questionnaire (the questionnaire composition, the summary of the questionnaire contents) and the contents of the questionnaire results need to be explained in detail.

2. Page 3, line 131

you explained the degree of paramedic, and it would be helpful for readers to understand if you also explained the scope of work and the composition of paramedic at fire stations and paramedics at hospitals.

Thank you.

Comments on the Quality of English Language

Minor editing of English language required.

Reviewer 2 Report

Comments and Suggestions for Authors

The authors concentrate their research on the PediSTAT application in a paediatric context, examining the incidence of prescription errors in the absence of this application. It is this author's opinion that this could be a viable method of presenting this application. However, given the existence of other available options (e.g.PEMSoft Moblie or general MDCalc ), the study's impact on health outcomes is limited.

The authors demonstrate the significance of incorrect drug selection, dosage, and concentration. However, it is essential that the authors elucidate the impact of these factors. It is important to note that the healthcare system in Saudi Arabia differs from that of other countries, particularly in Europe. Therefore, it would be beneficial to emphasise the professional activity ito better align it with the specific context.

The sentence ,lines 50-52, should be clarified.
The sample should emphasise the rationality and qualifications of the professionals. It is essential to differentiate between the groups in question. It would be beneficial to understand how the completion of the survey was monitored, given that it was carried out remotely. A detailed characterisation of the cases presented is also required. The authors should present some images of the application and a drawing summarising the study.
The discussion requires further development. Since other applications have not been used, it would be beneficial to discuss their existence and characteristics. The study has a very commercial design that should be avoided.

Reviewer 3 Report

Comments and Suggestions for Authors

There are several points in the manuscript that are not sufficiently explained.

Line 46:

Is 3.7% more or less than in adult patients?

Line 139:

What was the method used to allocate the intervention and control groups in a 1:1 ratio?

How was randomisation generated?

Line 145:

Was it possible to strictly control the control group so that they did not use such a smartphone application?

Results:

A flowchart of the study participants needs to be created.

Were there any statistically significant baseline differences between the control and intervention groups?

Line 206:

Even when using PediSTAT, one third of the answers seemed to be still wrong. If this is the real, then a significant amount of ME is occurring. Why were these people unable to answer correctly even when using PediSTAT?

Lines 278-280: ‘This has resulted in higher accuracy rates when using the PediSTAT application for each vignette compared to the conventional method.’ should be deleted because it has the same meaning as the previous sentence.

The results of written tests and actual behaviour differ. Even if they recognized the correct answers, writing errors might be occured. Even if they get the correct answer in a test, they may take the wrong action. This intervention may be effective in preventing ME due to lack of knowledge or calculation errors. However, it may not solve the above-mentioned types of ME.
